# Low Temperature Plasma Suppresses Lung Cancer Cells Growth via VEGF/VEGFR2/RAS/ERK Axis

**DOI:** 10.3390/molecules27185934

**Published:** 2022-09-13

**Authors:** Yuanyuan Zhou, Yan Zhang, Jie Bao, Jinwu Chen, Wencheng Song

**Affiliations:** 1Anhui Province Key Laboratory of Medical Physics and Technology, Institute of Health & Medical Technology, Hefei Institutes of Physical Science, Chinese Academy of Sciences, Hefei 230031, China; 2Hefei Cancer Hospital, Chinese Academy of Sciences, Hefei 230031, China; 3School of Life Science, Hefei Normal University, Hefei 230061, China; 4Collaborative Innovation Center of Radiation Medicine, Jiangsu Higher Education Institutions and School for Radiological and Interdisciplinary Sciences, Soochow University, Suzhou 215123, China

**Keywords:** low temperature plasma, lung cancer cell, ROS, RNS, cell death

## Abstract

Low temperature plasma (LTP) is a promising cancer therapy in clinical practice. In this study, dielectric barrier discharge plasma with helium gas was used to generate LTP. Significant increases in extracellular and intracellular reactive species were found in lung cancer cells (CALU-1 and SPC-A1) after LTP treatments. Cells viability and apoptosis assays demonstrated that LTP inhibited cells viability and induced cells death, respectively. Moreover, Western blotting revealed that the growth of CALU-1 cells was suppressed by LTP via the VEGF/VEGFR2/RAS/ERK axis for the first time. The results showed that LTP-induced ROS and RNS could inhibit the growth of lung cancer cells via VEGF/VEGFR2/RAS/ERK axis. These findings advance our understanding of the inhibitory mechanism of LTP on lung cancer and will facilitate its clinical application.

## 1. Introduction

Globally, cancer is one of the chief catastrophic diseases in the world, which severely impacts quality of life. In China, the incidence and mortality of cancer have increased annually, with lung cancer being the main cause of cancer-related death [1]. Non-small cell lung cancer (NSCLC), which accounts for about 85% of all lung cancer subtypes [2]. Radiotherapy is one of the effective treatment methods for NSCLC, and common radiotherapy methods include X-knife, γ-knife, and radiofrequency ablation (RFA) [3,4]. However, while radiotherapy induces the death of cancer cells, it also damages healthy cells, resulting in side effects such as vomiting and nausea [5]. In addition, radiation can even cause negative side effects such as pneumonia [6,7], acute esophagitis [8], heart disease [9], and skin burns [10]. Therefore, there is an urgent need to explore a new therapy for patients with cancer.

In recent years, low temperature plasma (LTP) has been developed as a novel method of radiotherapy [11]. Plasma is the fourth state of matter, which can be characterized as an ionized gas that is generally electrically neutral. LTP medicine is a novel interdisciplinary field that combines theoretical biophysics with clinical therapy. It is used widely in disinfection and sterilization, chronic wound healing (bone, skin, and nerve regeneration), cancer treatment, as well as in other applications [11,12,13], having a broad prospect of development. Furthermore, it is believed that the ability of LTP to kill the cancer cells effectively is due to the production of ROS and RNS. Thus, they can induce the redox reaction, resulting in cancer cell growth arrest as well as cell death [14,15,16].

The VEGF/VEGFR/RAS/ERK axis may control downstream signaling proteins for tumor cell growth, survival, migration, and invasion. Furthermore, treatments of targeting expression of the VEGF/VEGFR2/Ras/ERK axis have been shown to exert anti-tumor roles, including sophoridine [17,18], usnic acid [19], apatinib [20,21], dovitinib [22], sorafenib [23] and sulforaphene plus photodynamic treatment [24]. In this study, LTP was used to inhibit lung cancer cells via the VEGF/VEGFR2/RAS/ERK axis, which provides a new direction for radiotherapy to inhibit the growth of lung cancer cells.

## 2. Results and Discussion

### 2.1. LTP Impaired Survival Abilities of CALU-1 and SPC-A1 Cells

The morphology and survival rate of the lung cancer cells were significantly changed after LTP treatment, and phenomena of time-dependent cytotoxicity by LTP treatment in both CALU-1 and SPC-A1 cells. Figure 1a showed these changes by microscope capturing. After 24 h culture, lung cancer cells without LTP treatment could proliferate all over the bottom of petri dishes. With the extension of LTP treatment time, the number of attached cells decreased while the number of detached cells increased. There was no normal form of CALU-1 in the field of vision under the microscope after 60 s or longer treatment time, but for SPC-A1, some cells also remained in normal states in 75 s LTP treatment. Until 90 s LTP treatment, SPC-A1 could basically be observed out of its normal state. Therefore, CALU-1 was more sensitive to LTP than SPC-A1 cells.

In addition, we found that LTP significantly inhibited cell proliferation by comparing the CPI values of cells treated at various times in Figure 1b. The CPI for CALU-1, decreased to 96.73% (*p* = 0.0024), 85.98% (*p* < 0.001), 74.03% (*p* < 0.001), 68.15% (*p* < 0.001), 30.41% (*p* < 0.001), 20.68% (*p* < 0.001), respectively, of control after 15, 30, 45, 60, 75, and 90 s LTP treatment, while SPC-A1, 95.44% (*p* = 0.4068), 93.63% (*p* = 0.1625), 91.28% (*p* = 0.0415), 77.16% (*p* < 0.001), 66.97% (*p* < 0.001), 55.14% (*p* < 0.001), respectively. There was a significant difference of the CPI values between experimental and control groups in both CALU-1 and SPC-A1 (*p* < 0.001). Moreover, when treated with LTP for a shorter time (15 or 30 s), the inhibition in the two cell lines was not significantly different (*p* > 0.005). In contrast, CALU-1 was more sensitive to the killing effect than SPC-A1 after a longer LTP treatment time such as 45 s (*p* < 0.001), 60 s (*p* = 0.0141), 75 s (*p* < 0.001), and 90 s (*p* < 0.001). Therefore, LTP could impaired survival and proliferation abilities as well as damage cell morphology on CALU-1 and SPC-A1 cells efficiently.

As a radiotherapeutic method for cancer, LTP is more destructive to cancer cells than to normal tissue compared with other conventional chemotherapy [25,26]. Thus, LTP is promising method of reducing the pain in side effects caused by high radiotherapy doses or cancer recurrence caused by insufficient doses [27]. However, in lung cancer, this selectivity is somewhat “weak” [28]. Furthermore, there are few studies, [29,30] and the co-action of different types of radiotherapy, chemotherapy and LTP on tumor cells is unclear.

### 2.2. LTP Increased Extracellular ROS and RNS Levels

The production of reactive species was evaluated in the extracellular region, and Figure 2 shows the concentration of extracellular ROS and RNS generated by LTP. Under the same duration of LTP treatment, there was no significant difference between CALU-1 and SPC-A1.

Under the range of LTP exposure time, the concentration of ROS increased by 19.31% (*p* = 0.0195), 21.53% (*p* = 0.0097), 56.35% (*p* < 0.001), 67.74% (*p* < 0.001), 86.10% (*p* < 0.001) in 15, 30, 45, 60 and 75 s compared to 0 s LTP exposure, respectively. While the concentration of RNS was increased 3.78 (*p* < 0.005), 4.76 (*p* < 0.005), 6.53 (*p* < 0.001), 9.04 (*p* < 0.001), 11.04 (*p* < 0.001), and 13.79-fold (*p* < 0.001), respectively. Thus, we also found a time-dependent increase in the extracellular ROS and RNS. Moreover, reactive species in cells culture medium were significantly increased, which was consistent with the trend reported in other reasearch [31].

### 2.3. LTP Increased Intracellular ROS and RNS Generation

CALU-1 and SPC-A1 cells were cultured for 6 h after LTP treatment. Intracellular ROS was markedly increased in both CALU-1 and SPC-A1 cells in a time-dependent manner, as shown in Figure 3a,b. The green fluorescence probe DCF was observed by fluorescence microscopy. The intracellular ROS levels were low in both CALU-1 and SPC-A1 cells in the control groups, while the fluorescence intensity in the intracellular space increased 1.69-fold (*p* = 0.0494) and 3.51-fold (*p* = 0.0018) in CALU-1 cells, and 2.91-fold (*p* < 0.001) and 3.94-fold (*p* < 0.001) in SPC-A1 cells after 30 and 75 s LTP treatment, respectively.

An increase in intracellular RNS was also be observed by Figure 3c. The concentration of NO increased to 4.69% (*p* = 0.7153) and 36.56% (*p* = 0.0027) in CALU-1 cells, and to 40.49% (*p* = 0.0260) and 82.75% (*p* = 0.0008) in SPC-A1 cells after 30 and 70 s LTP exposure, respectively. There was a significant difference in the intracellular ROS and RNS levels between the experimental and control groups (*p* < 0.001). Therefore, both intracellular ROS and RNS contents increased after LTP treatment. Aside from physical factors, such as UV or heat, which exert a significant role in killing cancer cells [32], intracellular ROS and RNS have been recognized as effective tumor suppressors [33].

### 2.4. LTP Induced CALU-1 and SPC-A1 Cell Death

The proportion of live cells decreased from 88.17 to 7.53% (*p* < 0.001) for CALU-1, and from 95.1 to 48.8% (*p* = 0.0056) for SPC-A1 as the LTP treatment time increased from 0 s to 75 s after 24 h incubation, respectively (Figure 4). Apoptosis of LTP-treated SPC-A1 and CALU-1 cells was observed after 24 h. The main mechanism of LTP killing cancer cells is to induce cell apoptosis, pyroptosis, necrosis or other programmed cell deaths [34,35,36]. Since previous experimental results showed that CALU-1 was more sensitive to LTP than SPC-A1, subsequent experiments focused on CALU-1 cells.

### 2.5. LTP Resulted in Depolarization of the Mitochondrial Membrane Potential of CALU-1 Cells

The intensity of green fluorescence increased while that of red fluorescence decreased. This was because JC-1 mainly exists in the mitochondria of cells in the form of aggregates; but after LTP treatment, JC-1 became a monomer, as shown in Figure 5. Thus, LTP resulted in depolarization of the mitochondrial membrane potential of CALU-1 cells, which is characteristic of early apoptosis [37].

### 2.6. LTP Inhibited the Migration of CALU-1 Cells Migration

Carcinoma cells were detected by the scratch assay as shown in Figure 6. The ratio for 0, 30, 45, 60 s LTP treatment after 4 h incubation was decreased 92.99% (*p* = 0.8202), 69.47% (*p* = 0.0079), 34.86% (*p* = 0.0027), and after 16 h incubation it were 77.13% (*p* = 0.0237), 20.74% (*p* < 0.001), 6.3% (*p* < 0.001) compare with control group. The migration area ratio decreased by 96.81% (*p* > 0.9999), 36.72% (*p* = 0.0015), 11.90% (*p* < 0.001) after 30, 45, 60 s LTP treatment, respectively, as the incubation time increased from 4 h to 16 h. Meanwhile, there was a significant difference in the cell migration area between experimental and control groups in both 4 h and 16 h incubation after LTP treatment (*p* < 0.001).

The longer the LTP exposure treatment, the more obvious the inhibitory effect on cell migration for the same incubation time. Moreover, the longer the incubation time, the greater the inhibition of cell migration for the same LTP exposure time. Thus, LTP inhibited the migration of lung cancer cells in a time-dependent manner.

### 2.7. LTP Induced CALU-1 Cell Death via VEGF/VEGFR2/RAS/ERK Axis

To clearly understand the phenomenon of LTP causing lung cancer cell death, we continued to explore the relevant molecular mechanism. Thus, the expression of the VEGF/VEGFR2 axis and other correlative proteins of the downstream pathway were investigated in Figure 7. We found that the expression of VEGF, VEGFR2, RAS and ERK decreased significantly with the increase in LTP time. VEGF is involved in the synthesis of blood vessels by endothelial cells and is essential for growth and development. However, it is also widely expressed in diabetic complications (macular degeneration, atherosclerosis) [38,39], pulmonary edema [40], and tumor angiogenesis [41].

While VEGFR is the main functional receptor of VEGF, and Anti-VEGFR2 therapy can interfere with the AKT or ERK pathway and thus play an anti-tumor role [42]. Moreover, RAS regulates cell proliferation and migration, as well as activation, has been related to tumor angiogenesis [43]. RAS activation can also stimulate proteins, such as ERK, which are classified as proto-oncogenes [44], and is associated with abnormal activation of ERK in the occurrence of a variety of human cancers. Targeting the RAS / ERK pathway in cancer was a common target in cancer therapy [45]. Hence, LTP inhibited lung cancer cells by acting on the VEGF/VEGFR2/RAS/ERK axis in a time-dependent manner, as shown in Figure 8.

## 3. Materials and Methods 

### 3.1. NSCLC Cell Lines and Cultures

The human lung adenocarcinoma cell lines CALU-1 and SPC-A1 were purchased from the ATCC cell bank, and all of them were cultured in RPMI-1640 medium (GIBCO, Carlsbad, CA, USA) supplemented with 10% fetal bovine serum (LONSERA, Shanghai, China) and 1% penicillin/streptomycin (NCM Biotech, Suzhou, China). Cells were cultured in an incubator (Thermo Fisher Scientific, Waltham, MA, USA) under 37 °C and 5% CO_2_. Subsequently, cells were cultured as a monolayer growth and seeded in 60 mm Petri dishes (Sangon Biotech, Shanghai, China). Cells for experimental use were cultured to cover 80% of the area of the Petri dishes.

### 3.2. LTP Device

The atmospheric pressure DBD plasma device was designed and fabricated in our lab, which was schematically illustrated by Figure 9a,b. The dielectric barrier is about 1 mm thick quartz glass, which inserted into the discharge space. Our DBD device has a high-voltage electrode and a grounding electrode, and high energy density LTP was produced between the two electrodes. The capping and corresponding bases of the reactor fit each other completely were made of quartz materials. Four accommodating grooves were on the base, which can place 60 mm petri dishes. Additionally, there were four vent holes around the capping for working gas in and another one. When it is closed, capping and base can also completely cover the Petri dishes. In addition, pure helium (He, 99.999%) was used as the discharge gas, with a flow rate of 1 L/min. The DBD device ran for 90 s after He gas introduction to ensure reactors where He abounds before treating the cells. The LTP was generated under an effective voltage of 3.78 kV with a frequency of 25 kHz in Figure 9c. 

In order to ensure that the cells receive enough nutrition, fresh medium should be replaced by the old one before LTP treatment. Cells in RPMI-1640 were prepared to 5 mL in Petri dishes for 0, 15, 30, 45, 60, 75, and 90 s, and then cultured for 0, 6, 12, 24 h for use in various experiments.

### 3.3. Cell Viability Assay

The survival rate of cells treated with LTP was measured with 3-(4, 5-dimethylthiazol-2-yl)-2,5-diphenyltetr-azolium bromide (MTT, Sigma-Aldrich, St. Louis. MO, USA). CALU-1 and SPC-A1 cells were seeded in a 60 mm Petri dish at an initial concentration of 6 × 10^5^ cells/mL and cultured in 4 mL RPMI-1640 medium. Cells were grown to 70% confluence, exposed to the LTP apparatus for 0, 15, 30, 45, 60, 75, 90 s, and subsequently cultured for 24 h. MTT work solution (0.5 g/mL, total of 1.5 mL) was added to each Petri dish before discarding the medium. Following incubation 4 h, at 37 °C in the dark, the MTT solution was discarded and the same volume of DMSO (Sangon Biotech, Shanghai, China) was added to dissolve formazan in live cells. Then, 200 μL crystal violet dissolved in DMSO was transferred to a 96-well plate for testing in a microplate reader (Hiwell-Diatek, Wuxi, China). The absorbance was measured at a wavelength of 492 nm, and the cell proliferation index (CPI) was defined as described previously [46].
(1)$$\mathrm{CPI}=\frac{{\mathrm{OD}}_{\mathrm{value}\;\mathrm{of}\;\mathrm{treatment}}-{\mathrm{OD}}_{\mathrm{value}\;\mathrm{of}\;\mathrm{control}}}{{\mathrm{OD}}_{\mathrm{value}\;\mathrm{of}\;\mathrm{control}}}$$
where, the treatment represents the lung cells were treated by LTP, and the control represents non-treated lung cells.

### 3.4. Extracellular Reactive Species Detection

CALU-1 and SPC-A1 lung cancer cells were treated by LTP apparatus for 0, 15, 30, 45, 60, 75 s, and the levels of ROS in cell culture medium were detected at once by Hydrogen Peroxide Test Kit (Beyotime, Shanghai, China). Absorbance was measured at a wavelength of 560 nm for testing.

RNS in culture medium was evaluated using a Nitrite Test Kit (Jiancheng Bioengineering Institute, Nanjing, China), according to the manufacturer’s instructions. A colorimetric assay was used to evaluate the concentration of ROS and RNS in the extracellular area, and the contents in each sample were calculated from the standard curve.

### 3.5. Intracellular Reactive Species Detection

Lung cancer cells were treated by LTP apparatus for 0, 30, and 75 s, and then cultured for 6 h. Then, intracellular ROS formation was assessed by a Reactive Oxygen Species Assay Kit (Beyotime, Shanghai, China). The medium was discarded, cells were incubated with the probe for 20 min, and then cells were washed four times with PBS. Detection of DCF fluorescence can be used to determine the concentration of intracellular ROS. Green-fluorescent images were obtained by fluorescence-inverted microscopy (Olympus, Tokyo, Japan).

Intracellular RNS formation was assessed using a Nitric Oxide Test Kit (Beyotime, Shanghai, China), and Western & IP cell lysis buffer (Beyotime, Shanghai, China) was used to detect cellular disruption and intracellular RNS release. In 96-well plates, 50 μL cell lysates were added to each well (Griess assay reagents 1 and 2 in turn). The absorbance was measured at a wavelength of 520 nm and compared with a standard curve.

### 3.6. Flow Cytometer Analysis

CALU-1 and SPC-A1 cells were cultured in an incubator 24 h after LTP treated for 0, 30, 45, and 75 s. Then, the type of cell death was determined using an Annexin V-FITC Apoptosis Detection Kit (Beyotime, Shanghai, China). All of the samples were digested by trypsin without EDTA, collected by centrifugation in tubes, and then washed and resuspended in PBS or Binding Buffer. Later, cells were stained by both Annexin V-FITC and PI solutions for 8 min in the dark. Finally, each sample was analyzed using a CytoFlex Flow Cytometer (Accuri C6, Bedford, MA, USA). Normal cells were treated with 4% paraformaldehyde (Biosharp, Hefei, Anhui, China) at 55 °C water to induce apoptosis as a control for fluorescence compensation regulation, to remove overlap of the aging spectrum and to set the position of the gates. The green fluorescence for Annexin V-FITC was captured by FITC channel (FL-1), and the red fluorescence for PI by FL-3, respectively. Data were acquired using CytExpert Software 2.3.0, Beckman Coulter ( Brea, CA, USA).

### 3.7. JC-1 Detection

To investigate the effect of LTP on the mitochondrial membrane potential of lung cancer cells, cells were treated with LTP for 45 s and then incubated with JC-1 staining solution for 20 min before being discarded. Medium was added to the Petri dish, and changes in red and green fluorescence intensity were observed using a fluorescence-inverted microscope. When the mitochondrial membrane potential decreased, JC-1 changed from a polymer to a monomer, and the red fluorescence intensity became weak while the green fluorescence intensity increased.

### 3.8. Scratch Assay

Cells were cultured with serum-free medium, and 200 μL yellow sterile pipette tips were used to scratch a straight line on the bottom of the Petri dishes with uniform force. After washing with PBS three times, cells were treated with LTP for 0, 30, 45, and 60 s and cultured in an incubator for 4 and 16 h. Then, cells were observed with a microscope (Mshot, Guangzhou, China) for evaluating the degree of migration.

### 3.9. Western Blot Analysis

CCALU-1 cells were centrifugally collected after LTP treatment for 0, 30, 45, and 60 s and lysed on ice for 20 min in RIPA cell lysis buffer (Beyotime, Shanghai, China). After protein supernatants were extracted at 4 °C (CenLee, Hunan, China), the total protein concentration was determined by BCA protein quantification assay (BCA Protein Assay kit, Beyotime, China). The protein supernatants were added to loading buffer (Beyotime, Shanghai, China), denaturated at 100 °C for 20 min, and further cooled to room temperature. Proteins were separated by 10% SDS-PAGE gels, and electrophoresis was performed at 100 V for 120 min. The proteins in the gel were then transferred to the NC membrane (Beyotime, Shanghai, China), and the membrane was blocked with 5% non-fat milk in TBST for 2 h, and then incubated with primary antibodies at 4 °C overnight. Several primary antibodies were used, anti-VEGF (Cat. no. A12303), anti-VEGFR2 (Cat. no. A11127), anti-Ras (Cat. no. A4735), anti-ERK (Cat. no. A16686), anti-p-ERK (Cat. no. AP0485), and anti-Beta-Actin (Cat. no. AC026) (ABclonal, Wuhan, China). Equal loading was confirmed using the loading control anti-Beta-actin. The NC membranes were incubated for 40 min at room temperature with the relevant second antibody, and the dilution ratio of HRP-conjugated secondary antibody was 1:10,000. Finally, ECL Plus Reagent (Thermo Fisher Scientific, Waltham, MA, USA) was added, and the quantification of protein bands were visualized using a Chemiluminescence Gel Imaging System (Tanon, Shanghai, China).

### 3.10. Statistical Analysis

The data of assays were launched at least three repeating independent experiments (*n* = 3) and presented as means ± S.D. Data were determined using one- or two-way analysis of variance (ANOVA). A *p*-value < 0.05 (*) was defined as statistically significant.

## 4. Conclusions

In conclusion, extracellular and intracellular reactive species were dependent on the time of LTP treatment. MTT assay showed that cell viability was significantly reduced after 24 h incubation of LTP treatment, and apoptosis experiments showed an increase in lung cancer cell death after 6 h of LTP treatment. Additionally, Western Blot assay results demonstrated ROS and RNS produced by LTP inhibited lung cancer CALU-1 cells via VEGF/VEGFR2/RAS/ERK axis. Thus, the results reveal the mechanism of LTP killing lung cancer cells, laying the foundation for large-scale clinical application of LTP in the future.

## Figures and Tables

**Figure 1 molecules-27-05934-f001:**
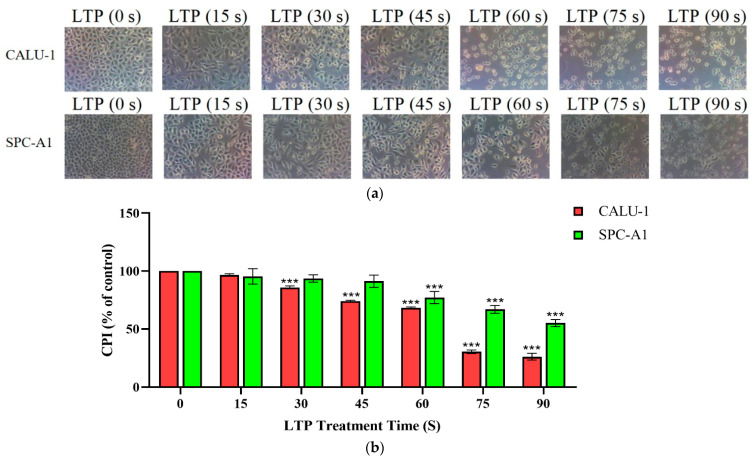
LTP impaired the survival ability of CALU-1 and SPC-A1 cells. (**a**) CALU-1 and SPC-A1 cells were detected under a light microscope 1 day after treatment with LTP for different times; (**b**) Comparison of the CPI for CALU-1 and SPC-A1 at 1 day after treatment with LTP for different times. Data represent the mean ± SD of three independent experiments. *** *p* < 0.001; compared with the 0 s LTP treatment time group.

**Figure 2 molecules-27-05934-f002:**
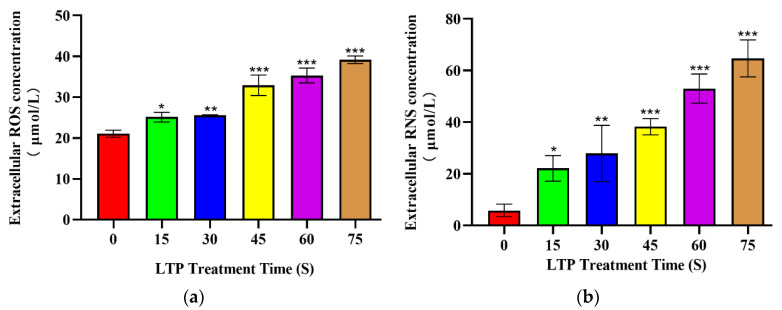
LTP increased extracellular ROS and RNS levels. Production of (**a**) ROS (**b**) RNS within the culture media compared to the control group immediately after treatment with LTP for 15, 30, 45, 60, 75, 90 s. Data represent the means ± SD of three independent experiments. * *p* < 0.05; ** *p* < 0.01; *** *p* < 0.001; compared with the 0 s LTP exposure group.

**Figure 3 molecules-27-05934-f003:**
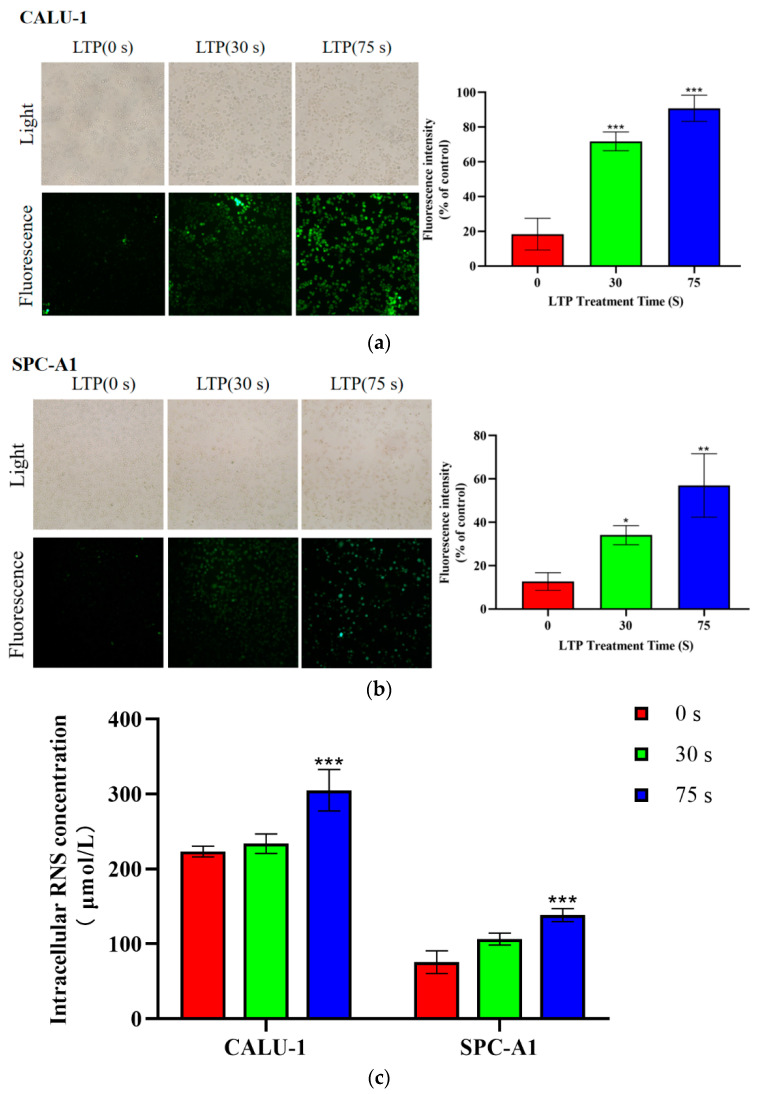
LTP induced the generation of intracellular ROS and RNS. Production of (**a**,**b**) ROS and (**c**) RNS within CALU-1 and SPC-A1 cells compared to the control group following treatment with LTP for 0, 30, and 75 s, and 6 h incubation. Data represent the means ± SD of three independent experiments. * *p* < 0.05; ** *p* < 0.01; *** *p* < 0.001; compared with the 0 s LTP treatment group.

**Figure 4 molecules-27-05934-f004:**
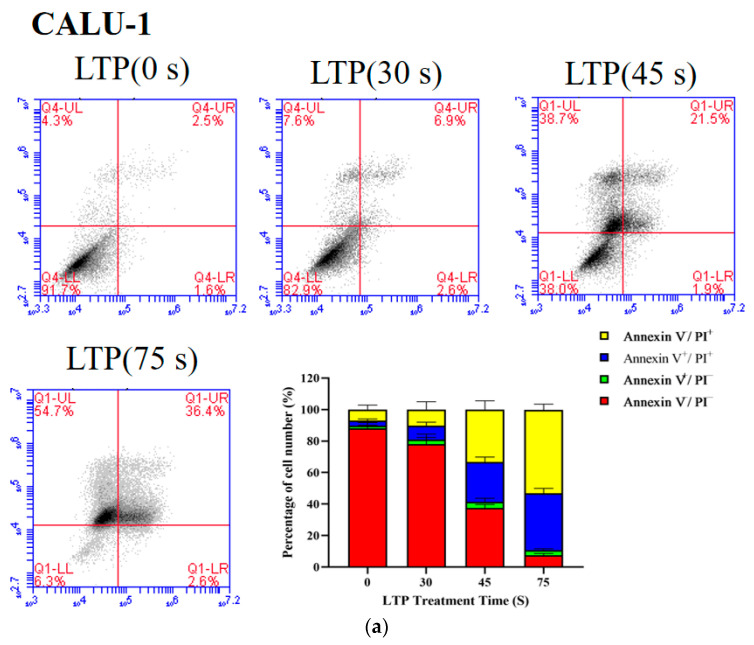
LTP induced the death of CALU-1 and SPC-A1 cells. Flow cytometry was performed to analyze cell death by PI and Annexin V-FITC staining in (**a**) CALU-1 and (**b**) SPC-A1 cell lines after LTP treatment for 30, 45, and 75 s, and measured after 24 h.

**Figure 5 molecules-27-05934-f005:**
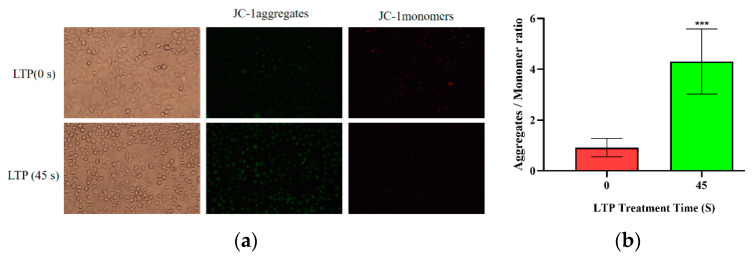
LTP resulted in depolarization of the mitochondrial membrane potential of CALU-1 cells. (**a**) Fluorescence microscope was performed to detect JC-1 monomers and aggregates and (**b**) the data were quantitated and graphed (paired-samples *t* test). *** *p* < 0.001; compared with the 0 s LTP treatment group.

**Figure 6 molecules-27-05934-f006:**
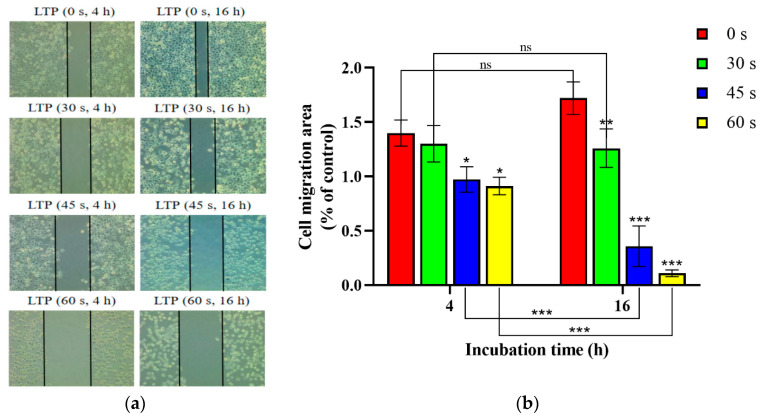
LTP affects the cell migration. (**a**) The relative ratio was calculated as the ratio between the experimental groups and the 0 s LTP treatment groups of CALU-1 cells at 0, 30, 45, and 60 s, with 4 and 16 h thermostatic incubation. (**b**) The data were quantitated and graphed. * *p* < 0.05; ** *p* < 0.01; *** *p* < 0.001; compared with the 0 s LTP treatment group.

**Figure 7 molecules-27-05934-f007:**
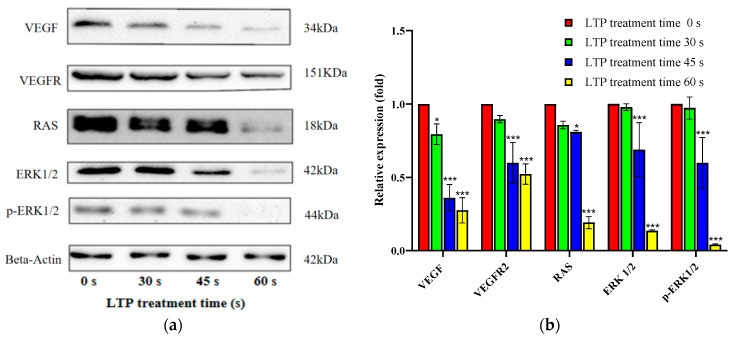
LTP induced CALU-1 cell death via VEGF/VEGFR2/RAS/ERK axis. (**a**) The expressions of key proteins in this signaling pathway were assessed by Western Blot analysis, (**b**) the data were quantitated and graphed. Data represent the means ± SD of three independent experiments, compared with 0 s LTP treatment time group. * *p* < 0.05; *** *p* < 0.001; compared with the 0 s LTP treatment group.

**Figure 8 molecules-27-05934-f008:**
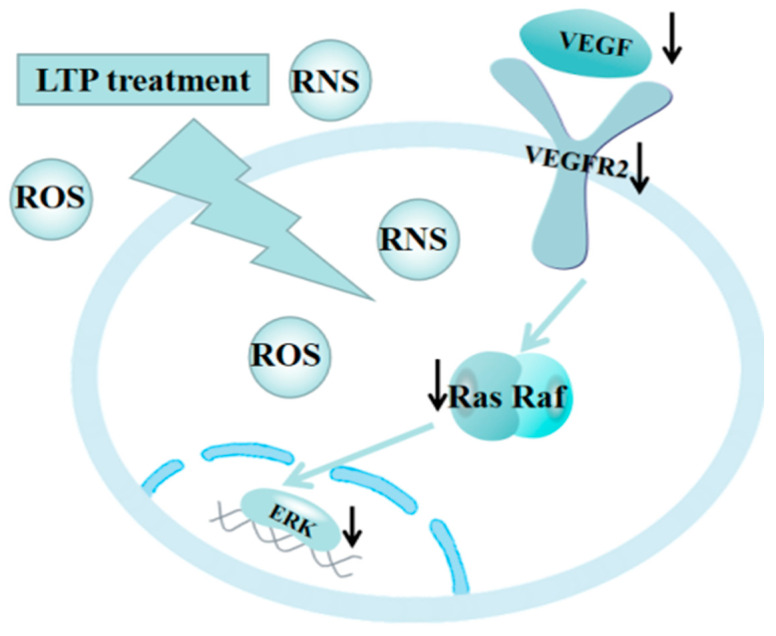
LTP can generate ROS and RNS and suppress the growth of lung cancer cells via inhibition of the VEGF/VEGFR2/RAS/ERK axis. Black arrows in the figure denote proteins with reduced expression following LTP treatment.

**Figure 9 molecules-27-05934-f009:**
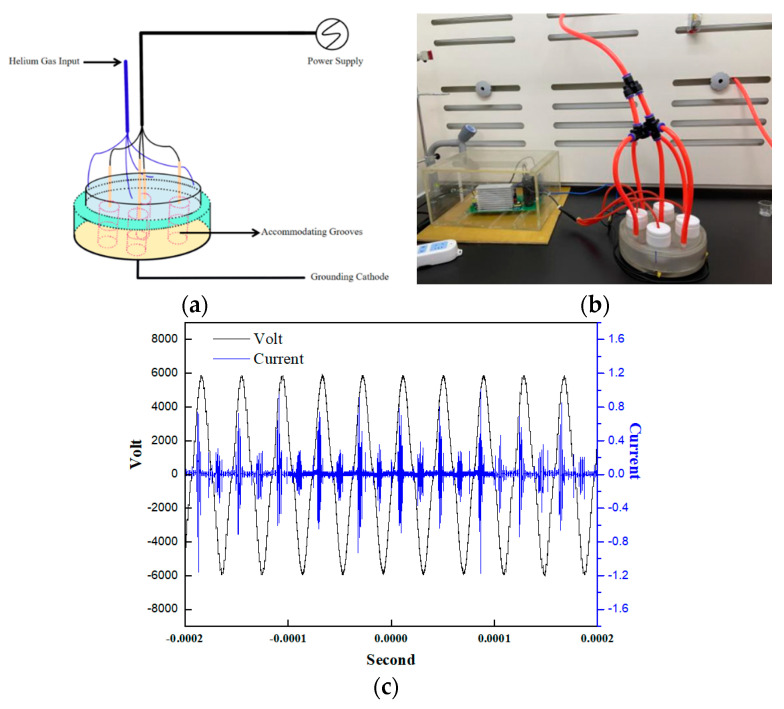
DBD LTP apparatus. (**a**,**b**) Schematic and images pictures of the DBD LTP apparatus. (**c**) Voltage and current waveforms of the LTP apparatus.

## Data Availability

Not applicable.

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
