# Peer review of "Low Temperature Plasma Suppresses Lung Cancer Cells Growth via VEGF/VEGFR2/RAS/ERK Axis"

_molecules, 2022, doi:10.3390/molecules27185934_

Round 1

Reviewer 1 Report

The manuscript entitled: "Low temperature plasma suppresses lung cancer cells growth via VEGF / VEGFR2 / RAS / ERK axis" reports the effective use of LTP against lung cancer cells. The authors have also investigated the molecular aspects of mechanistic details. 

Overall, the study is exciting. However, there are several significant concerns I believe authors should consider to improve the quality of the manuscript.  

Few sentences are unnecessarily long. For instance, the first statement of the abstract is four lines, Line #12 - Line #15. I strongly advise authors to make these kinds of statements short without compromising the meaning throughout the manuscript.

Line #29: what do authors mean by cancer cottoms?

Line #31: RFA full form?

Nausea and vomiting are common side effects of almost all medicines. Radiotherapy has much worse side effects. Adding a line of that after line #32 will better justify the statement in Line #33 (Therefore, it is urgent to explore......).  

Method 2.3 needs a more detailed description. How many cells were seeded? The plate format in which cells were seeded? Media volume? All these details should be mentioned. The authors mention that 1.5 mL MTT was added to the petri dish and incubated for 4 hours, then discarded MTT solution and an equal amount (that will be 1.5 mL) of DMSO was added to dissolve crystals and transferred to 96-well plate. How did the authors manage to add 1.5 mL to 96 well plates? It is not clear. I encourage authors to rewrite this whole section.

Methods 2.9: No blocking step? Adding catalog numbers for the antibodies will be helpful for readers. Also, Anti-ATCB should be anti- Beta-Actin

Line #163: ......Gel Imaging System were used for Protein quantitative???? what does it mean?  

Figure #6: the fluorescence images are not clear. A better quality image will be helpful. Additionally, it is essential to mention the molecular weights of all the markers tested. 

Figure 8: Beta-Actin intensity also appears to reduce as time increases. What software was used to perform the densitometric analysis? Please provide the original image of blots of three independent experiments as supplementary material.  

The manuscript has several grammatical errors and needs extensive English editing.

Author Response

Comments from the editors and reviewers:

Reviewer: 1

  1. Few sentences are unnecessarily long. For instance, the first statement of the abstract is four lines, Line #12 - Line #15. I strongly advise authors to make these kinds of statements short without compromising the meaning throughout the manuscript.

Reply: Thanks a lot for the reviewer’s comments. According to the reviewer’s suggestion, we revised the abstract in revised manuscript. Please see Page 1 Line 12-21.

Revised parts:

Low temperature plasma (LTP) is a promising cancer therapy in clinical practice. In this study, dielectric barrier discharge plasma with helium gas was used to generate LTP. Significant increases in extracellular and intracellular reactive species were found in lung cancer cells (CALU-1 and SPC-A1) after LTP treatments. Cells viability and apoptosis assays demonstrated that LTP inhibited cells viability and induced cells death, respectively. Besides, Western blotting revealed that the growth of CALU-1 cells was suppressed by LTP via the VEGF / VEGFR2 / RAS / ERK axis for the first time. The results showed that LTP-induced ROS and RNS could inhibit the growth of lung cancer cells via VEGF / VEGFR2 / RAS / ERK axis. These findings advance our understanding of the inhibitory mechanism of LTP on lung cancer and will facilitate its clinical application. 

  1. Line #29: what do authors mean by cancer cottoms?

Reply: Thanks a lot for the reviewer’s comments, cancer subtypes was incorrectly translated as cancer cottoms in a translation software. We are very sorry for using the wrong expression without checking it carefully. After your kindly reminding, we found this translation was not suitable, so now it has been corrected. Please see Page 1 Line 29.

Revised parts:

Non-small cell lung cancer (NSCLC), which accounts for about 85% of all lung cancer subtypes [2].

  1. Line #31: RFA full form?

Reply: We greatly appreciate the reviewer’s detailed comments. RFA full form is radiofrequency ablation. This word was included in this sentence. Please see Page 1 Line 30.

Revised parts: X-knife, γ-knife, radiofrequency ablation (RFA) [3,4].

  1. Nausea and vomiting are common side effectsof almost all medicines. Radiotherapy has much worse side effects. Adding a line of that after line #32 will better justify the statement in Line #33 (Therefore, it is urgent to explore......).  

Reply: Thanks a lot for the reviewer’s comments. According to the reviewer’s requirements, we have added the more details on the side effects of radiotherapy. Please see Page 1 Line 32-35.

Revised parts:

In addition, radiation can even cause negative side effects such as pneumonia [6, 7], acute esophagitis [8], heart disease [9], and skin burns [10]. Therefore, it is an urgent to explore a new therapy for patients with cancer.

  • Song, Y.H., Chai, Q., Wang, N.L., Yang, F.F., Wang, G.H., Hu J.Y. X-rays induced IL-8 production in lung cancer cells via p38/MAPK and NF-kappa B pathway. International Journal of Radiation Biology, 202096: 1374-1381.
  • Okumus, D., Sarihan, S., Gozcu, S., Sigirli, D.The relationship between dosimetric factors, side effects, and survival in patients with non-small cell lung cancer treated with definitive radiotherapy. Medical Dosimetry, 2017 42: 169-176.
  • Xiong, W.J., Xu, Q.F., Xu, Y., Sun, C.J., Li, N., Zhou, L., Liu, Y.M., Zhou, X.J., Wang, Y.S., Wang, J., Bai, S., You, L., Gong, Y.L. Stereotactic body radiation therapy for post-pulmonary lobectomy isolated lung metastasis of thoracic tumor: survival and side effects. Bmc Cancer, 201414: 719.
  • Ming, X., Feng, Y.M., Yang, C.W., Wang, W., Wang, P., Deng, J. Radiation-induced heart disease in lung cancer radiotherapy: A dosimetric update. Medicine, 201695: e5051.
  • Xie, Y., Zhang, H., Hao, J.F., Zhao, W.P., Wu, Z.H., Qiu, R., Wang, X.H. Effect of p53 on lung carcinoma cells irradiated by carbon ions or X-rays. Nuclear Science and Techniques, 200920: 146-151.

  1. Method 2.3 needs a more detailed description. How many cells were seeded? The plate format in which cells were seeded? Media volume? All these details should be mentioned. The authors mention that 1.5 mL MTT was added to the petri dish and incubated for 4 hours, then discarded MTT solution and an equal amount (that will be 1.5 mL) of DMSO was added to dissolve crystals and transferred to 96-well plate. How did the authors manage to add 1.5 mL to 96 well plates? It is not clear. I encourage authors to rewrite this whole section.

Reply: Thanks for reviewer’s kindly and thorough suggestion. 3×106 cells (cell density at 6×105 cells / mL) were seeded in 60 mm Petri dish, and Media volume was 5 mL. And actually, we take 200μL out of 1.5 mL DMSO adding to 96 well plates. As advised, we rewrite this section, Please see Page 3 Line 89-99. 

Revised parts:

The survival rate of cells treated with LTP was measured with 3-(4, 5-dimethylthiazol-2-yl)-2,5-diphenyltetr-azolium bromide (MTT, Sigma-Aldrich, USA). CALU-1 and SPC-A1 cells were seeded in a 60 mm Petri dish at an initial concentration of 6 × 105 cells / mL, and cultured in 4 mL RPMI-1640 medium. Cells were grown to 70% confluence, exposed to the LTP apparatus for 0, 15, 30, 45, 60, 75, 90 s, and subsequently cultured for 24 h. MTT work solution (0.5 g/mL, total of 1.5 mL) was added to each Petri dish before discarding the medium. Following incubation 4 h, at 37℃ in the dark, the MTT solution was discarded and the same volume of DMSO (Sangon Biotech, Shanghai, China) was added to dissolve formazan in live cells. Then, 200 μL crystal violet dissolved in DMSO was transferred to a 96-well plate for testing in a microplate reader (Hiwell-Diatek, Wuxi, China). The absorbance was measured at a wavelength of 492 nm, and the cell proliferation index (CPI) was defined as described by previously [25].

  1. Methods 2.9: Noblocking step? Adding catalog numbers for the antibodies will be helpful for readers. Also, Anti-ATCB should be anti- Beta-Actin.

Reply: Thanks very much for reviewer’s meticulous review. We are very sorry for missing blocking step, and now we’ve added this in the revised manuscript. Please see Page 5 Line 172-173. Besides, catalog numbers for the antibodies was added and anti-ATCB was also corrected into anti-Beta-Actin in the revised manuscript according to the suggestion. Please see Page 5 Line 174-177. 

Revised parts:

Line 172-173:The proteins in gel were then transferred to the NC membrane (Beyotime, Shanghai, China), and the membrane was blocked with 5% non-fat milk in TBST for 2 h, and then incubated with primary antibodies at 4℃ overnight.

Line 174-177:Several primary antibodies were used, anti-VEGF (Cat. no. A12303), anti-VEGFR2 (Cat. no. A11127), anti-Ras (Cat. no. A4735), anti-ERK (Cat. no. A16686), anti-p-ERK (Cat. no. AP0485), and anti-Beta-Actin (Cat. no. AC026) (ABclonal, Wuhan, China). Equal loading was confirmed using the loading control anti-Beta-actin.)

  1. Line #163: ......Gel Imaging System were used for Protein quantitative???? what does it mean?  

Reply: We thank for the kindly comments of reviewer, which is very important for us to improve our manuscript. Accurately, this part are not described in detail. What we want to express is that the quantification of protein bands requires the use of a gel imaging system, which is visual. We corrected it in the revised manuscript. Please see Page 5 Line 181-182. 

Revised parts:

Lastly, ECL Plus Reagent (Thermo Fisher Scientific, Waltham, MA, USA) was added, and the quantification of protein bands were visualized using a Chemiluminescence Gel Imaging System (Tanon, Shanghai, China).

  1. Figure #6: the fluorescence images are not clear. A better quality image will be helpful. Additionally, it is essential to mention the molecular weights of all the markers tested.

Reply: We greatly appreciate the reviewer’s comment. According to the suggestions, the unclear fluorescence pictures in the manuscript have been replaced with a clearer pictures in Figure 4 (Page 8 Line 277-280). In the revised manuscript, the molecular weights of all the markers of protein have been added in Figure 8 (Page 11 Line 345). The revised content is as follows.

Revised parts:

(a) 

(b)

Figure 4

Figure 8.

  1. Figure 8: Beta-Actin intensity also appears to reduce as time increases. What software was used to perform the densitometric analysis? Please provide the original image of blots of three independent experiments as supplementary material.  

Reply: We greatly appreciate the reviewer for a meticulous review of this manuscript. Image J software was used to analyze the gray value of protein bands (Table 1). Then, we analyze outcomes (one-way ANOVA) that there was no significant difference between them (P = 0.9994、0.7647、0.8922). Therefore, we believe that the expression of Beta-Actin can be used as a protein reference.

Table 1. Gray value of protein bands

Group

Treatment time (s)

0 s

30 s

45 s

60 s

1

22254.36

24325.95

25650.82

22818.19

2

30499.65

27060

29894.55

27708.14

3

27089.58

27505.41

32658.12

35245.75

  1. The manuscript has several grammatical errors and needs extensive English editing.

Reply: Thanks for reviewer’s kindly comments. We corrected punctuation or grammatical errors and improved written expression the manuscript.

Page 1 Line 25-26: Globally, cancer is one of the chief catastrophic diseases in the world, which severely impacts quality of life.

Page 1 Line 26-27: In China, the incidence and mortality of cancer have increased annually, with lung cancer being the main cause of cancer-related death [1].

Page 1 Line 31-32: However, while radiotherapy induces the death of cancer cells, it also damages healthy cells, resulting in ...

Page 1 Line 36: In recent years, low temperature plasma (LTP) has been ...

Page 1 Line 38: Plasma is the fourth state of matter, which can be characterized as ...

Page 1 Line 39: LTP medicine is a novel interdisciplinary

Page 1 Line 46: The VEGF / VEGFR / RAS / ERK axis may control downstream signaling proteins ...

Page 2 Line 47-49: Furthermore, treatments of targeting expression of the VEGF / VEGFR2 / Ras / ERK axis have been shown to exert anti-tumor roles, including ...and sulforaphene plus photodynamic treatment.

Page 2 Line 52-53: In this study, ... which provides a new direction for radiotherapy to inhibit the growth of lung cancer cells.

Page 2 Line 60-64: Cells were cultured in an incubator (Thermo Fisher Scientific, Waltham, MA, USA) under 37℃ and 5% CO2. Subsequently, cells were cultured as a monolayer growth and seeded in 60 mm Petri dishes (Sangon Biotech, Shanghai, China). Cells for experimental use were cultured to cover 80% of the area of the Petri dishes.

Page 2 Line 76-77: The DBD device ran for 90 s after He gas introduction to ensure reactors where He abounds before treating the cells.

Page 3 Line 80-81: Schematic and images pictures...

Page 3 Line 85-86: In order to ensure that the cells get enough nutrition, fresh medium should be replaced by the old one before LTP treatment.

Page 3 Line 108-109: RNS in culture medium was evaluated using ..., according to the manufacturer’s instructions.

Page 4 Line 117-118: The medium was discarded, cells were incubated with the probe for 20 min, and then cells were washed four times with PBS.

Page 4 Line 130: CALU-1 and SPC-A1 cells were cultured in an incubator 24 h after LTP treated for 0, 30, 45, and 75 s.

Page 4 Line 145-147: To investigate the effect of LTP on the mitochondrial membrane potential of lung cancer cells, cells were treated with LTP for 45 s and then incubated with JC-1 staining solution for 20 min before being discarded.

Page 6 Line 222: As a radiotherapeutic... 

Page 6 Line 230-233: The production of reactive species was evaluated in the extracellular region, and Figure 3 shows the concentration of extracellular ROS and RNS generated by LTP. Under the same duration of LTP treatment, there was no significant difference between CALU-1 and SPC-A1.

Page 8 Line 279: Figure 4. LTP induced the generation of ... following treatment with LTP .., and ... 

Page 10 Line 303: which is characteristic of early apoptosis [38].

Reviewer 2 Report

The paper presented by Song et al. is surely of interest, because it investigated the mechanism of low temperature plasma on lung cancer and its clinical application.

The manuscript is well structured, and the methods are adequately described. I have only some minor request on specific topics.

1. How low temperature plasma effects alveolar epithelial cells or bronchial epithelial cells, there’s no control in some experiments.

2. Since there’s crosstalk between tumor cells and microenvironment, how low temperature plasma effects macrophages and lymphocytes.

3. The authors presented that low temperature plasma increased extracellular ROS and RNS levels, and the expression of VEGF, VEGFR2, RAS and ERK decreased significantly with the increase of LTP time. Then, how ROS and RNS effect the VEGF / VEGFR2 / RAS / ERK axis.

4. The authors wrote: “Therefore, the future research direction can with an eye towards the synergistic effect of chemosensitizers and LTP, or the combination therapy between LTP and immune therapy, in the hope that plasma can play a greater role in the clinical efficacy of lung cancer”. So far, there are few studies, and the co-action of different types of radiotherapy, chemotherapy and LTP on tumor cells is unclear.

5. In the present study, the inhibitory mechanism of LTP on lung cancer is investigated based on in vitro experiments, such as cell experiments. There is lack of reliable in vivo experiments.

Author Response

Reviewer: 2

  1. How low temperature plasma effects alveolar epithelial cells or bronchial epithelial cells, there’s no control in some experiments.

Reply: Thanks for reviewer’s kindly comments. In our experiment, the comparison of normal cells was not involved, because some scientists have proved that LTP could inhibit the viability of lung cancer cells while slightly impairs the viability of normal cells, such as bronchial cell BEAS-2B [1, 2]. Besides, LTP could also lead to a decrease the number of attached cells, and while no change in normal human Bronchial epithelial (NHBE) numbers under same treatments [3]. It is generally believed that the basic ROS in cancer cells is higher than that in normal cells, so when intracellular ROS rised by LTP, and thus ROS in cancer cells may reach the death threshold easily [4].

  • Kim,S.J., Chung, T. Cold atmospheric plasma jet-generated RONS and their selective effects on normal and carcinoma cells. Scientific Reports, 2016 6: 20332.
  • Kim,S.J., Joh, H.M., Chung, T. Production of intracellular reactive oxygen species and change of cell viability induced by atmospheric pressure plasma in normal and cancer cells. Appl Phys Lett, 2013 103: 023702.
  • Keidar,M., Walk, R., Shashurin, A., Srinivasan, P., Sandler, A., Dasgupta, S., Ravi, R., Preston, R.G., Trink, B. Cold plasma selectivity and the possibility of a paradigm shift in cancer therapy. Br J Cancer, 2011 105: 1295-1
  • Yan,D.Y., Sherman, J.H., Keidar, Cold atmospheric plasma, a novel promising anti-cancer treatment modality. Oncotarget, 2017 8: 15977-15995. 

  1. Since there’s crosstalk between tumor cells and microenvironment, how low temperature plasma effects macrophages and lymphocytes.

Reply: Thanks a lot for the reviewer’s comments. The important comments are very helpful for us to significantly improve the quality of the manuscript.

LTP leads to cancer cells release damage-related molecular patterns (DAMPs), such as Surface-to-exposed calreticulin (ECto-CRT) [1], heat shock proteins [2], ATP, and HMGB1 [3], which induced the ICD. CRT mediates macrophage phagocytosis of tumor cells, ATP promotes NK cells cytotoxicity, which caused specific T cell response, and induced effctive anti-tumor immune response [4]. In addition, ATP also promoted the secretion of cytokines, such as tumor necrosis factor A (TNFa), which could further improve the anti-tumor ability of macrophages [3].

  • Lin,A., Truong, B., Patel, S., Kaushik, N., Choi, E.H., Fridman, G., Fridman, A.,  Miller, V. Nanosecond-Pulsed DBD Plasma-Generated Reactive Oxygen Species Trigger Immunogenic Cell Death in A549 Lung Carcinoma Cells through Intracellular Oxidative Stress. Int J Mol Sci, 2017 18: 966.
  • Tabuchi,Y., Uchiyama, H., Zhao, Q.L., Yunoki, T., Andocs, G., Nojima, N., Takeda, K., Ishikawa, K., Hori, M., Kondo, T. Effects of nitrogen on the apoptosis of and changes in gene expression in human lymphoma U937 cells exposed to argon-based cold atmospheric pressure plasma. Int J Mol Med, 2016 37: 1706-17
  • Lin,A., Truong, B., Pappas, A., Kirifides, L., Oubarri, A., Chen, S.Y., Lin, S. J., Dobrynin, D., Fridman, G., Fridman, A., Sang, N., Miller, V. Uniform Nanosecond Pulsed Dielectric Barrier Discharge Plasma Enhances Anti-Tumor Effects by Induction of Immunogenic Cell Death in Tumors and Stimulation of Macrophages. Plasma Processes and Polymers, 2015 12: 1392-139
  • Chao M P, Jaiswal S, WeissmanT.R., Alizadeh, A.A., Gentles, A. J., Volkmer, J., Weiskopf, K., Willingham, S., Raveh, T., Park, C.Y., Majeti, R., Weissman, I.L. Calreticulin Is the Dominant Pro-Phagocytic Signal on Multiple Human Cancers and Is Counterbalanced by CD47. Science Translational Medicine, 2010 2: 63ra94.

  1. The authors presented that low temperature plasma increased extracellular ROS and RNS levels, and the expression of VEGF, VEGFR2, RAS and ERK decreased significantly with the increase of LTP time. Then, how ROS and RNS effect the VEGF / VEGFR2 / RAS / ERK axis.

Reply: Thanks a lot for the reviewer’s detailed and thorough comments. Considerable literature has grown up relating to ROS and RNS affect the cell status by change the expression of VEGF / VEGFR2 / RAS / ERK axis. Kaempferol triggered human umbilical vein endothelial cells (HUVECs) apoptosis through ROS-mediated VEGF / VEGFR2 / RAS / ERK expression decreases [1]. Perillyl alcohol (PA) could increase ROS in lung cancer cell line A549, which lead to the decrease of VEGF expression and cell apoptosis [2]. Pogostemon cablin essential oils (PPa extract) also induced the rise of ROS, which in turn lead to cell cycle arrest and cell apoptosis by VEGF expression decreased [3]. Besides, RNS caused DNA breakage and activated the DNA base repair enzyme PARP-1, and then inhibit VEGF/VEGFR2, resulting in cell apoptosis [4]. And RNS catalytic scavenger MnTE-2-PyP5+, a kind of peroxynitrite, lead to a decreases in VEGF to suppress breast cancer in mice [5].

  • Chin, H.K., Horng, C.T., Liu, Y.S., Lu, C.C., Su, C.Y., Chen, P.S., Chui, H.Y., Tsai, F.J., Shieh, P.C., Yang, S.J. Kaempferol inhibits angiogenic ability by targeting VEGF receptor-2 and downregulating the PI3K/AKT, MEK and ERK pathways in VEGF-stimulated human umbilical vein endothelial cells. Oncology Reports,2018, 39: 2351-2357.
  • Liu,X.R., Bai, Y.F., Liang, L.L., Feng, J., Deng, F. Anti-lung cancer effect and anti-angiogenesis therapy study of perillyl alcohol. Chinese Journal of Immunology, 2017 33: 859-8
  • Huang, X.F., Sheu, G.T., Chang, K.F., Huang, Y.C., Hung, P.H., Tsai, M.N. Pogostemon cablin Triggered ROS-Induced DNA Damage to Arrest Cell Cycle Progression and Induce Apoptosis on Human Hepatocellular Carcinoma In Vitro and In Vivo. Molecules, 202025: 5639.
  • Mathews, M.T., Berk,B. PARP-1 inhibition prevents oxidative and nitrosative stress-induced endothelial cell death via transactivation of the VEGF receptor 2 [J]. Arteriosclerosis Thrombosis and Vascular Biology, 2008, 28: 711-717.
  • Rabbani Z N, Spasojevic I, Zhang XW, Moeller B J, Haberle S, Vivar V J, Dewhirst M W, Vujaskovic Z, Haberle I B. Antiangiogenic action of redox-modulating Mn(III) meso-tetrakis (N-ethylpyridinium-2-yl) porphyrin, MnTE-2-PyP5+, via suppression of oxidative stress in a mouse model of breast tumor [J]. Free Radical Biology and Medicine, 2009, 47: 992-1004.

  1. The authors wrote: “Therefore, the future research direction can with an eye towards the synergistic effect of chemosensitizers and LTP, or the combination therapy between LTP and immune therapy, in the hope that plasma can play a greater role in the clinical efficacy of lung cancer”. So far, there are few studies, and the co-action of different types of radiotherapy, chemotherapy and LTP on tumor cells is unclear.

Reply: Thanks for reviewer’s kindly detailed comment. According to your suggestion, we added this vital describe to the manuscript in the corresponding place. Please see Page 6 Line 218-223.

Revised parts:

Besides, there are few studies, [30, 31] and the co-action of different types of radiotherapy, chemotherapy and LTP on tumor cells is unclear. 

  1. In the present study, the inhibitory mechanism of LTP on lung cancer is investigated based on in vitro experiments, such as cell experiments. There is lack of reliable in vivo experiments.

Reply: Thanks for reviewer’s kindly comment and suggestion, which is very important for us to improve our manuscript. Unfortunately, we can't carry out this experiment in a short period of time because there are no experimental conditions in our laboratory. But as the experimental process goes further in the future, we’ll actively carry out relevant animal experiments. Besides, we investigate the studies on LTP on lung cancer in vivo as follows.

Investigations on LTP effects lung cancer cells within 3D collagen models showed that as the depth of the tumor increases, the influence of plasma diminished [1]. Therefore, there are two main treatment types of in vivo studies on the anti-lung cancer effects of LTP: direct treatment, which needs to be combined with other therapies, or indirect treatment, which means LTP-activated liquid was injected into mice. Scientists proved that LTP could significantly inhibit the volume growth of transplanted NSCLC tumors in vivo study [2-5]. For direct, Li used LTP irradiated directily and iron oxide-based magnetic nanoparticles for synergetic to inhibit the growth of lung cancer in nude mice [3]. Besides, as the result of the limited size of LTP device, it is not possible to use LTP device to target internal organs directly. Therefore, Kim[4] invented a treatment system which combining microplasma and endoscope, hoping that it will be possible to use LTP to directly treat lung cancer in the future. For indirect, Song[5] prepared oral administration of plasma activation solution, and demonstrated that it could inhibit the growth of chemoresistant non-small cell lung cancer mice transplanted tumor in rats.

  • Karki S B, Gupta T T, YildirimA E, Kathryn M E, Halim A. Investigation of non-thermal plasma effects on lung cancer cells within 3D collagen matrices. Journal of Physics D-Applied Physics, 2017 50: 315401.
  • Li WT, Yu H L, Ding D J, Chen Z T, Wang S S, Li X J, Michael K, Zhang W F. Cold atmospheric plasma and iron oxide-based magnetic nanoparticles for synergetic lung cancer therapy. Free Radic Biol Med, 2019 130: 71-81.
  • Wang YH, Mang X Y, Li X, Tan F. Cold atmospheric plasma induces apoptosis in human colon and lung cancer cells through modulating mitochondrial pathway. Front Cell Dev Biol, 2022 10: 915785.
  • Kim J Y, Ballato J, Foy P, Hawkins T, Wei Y Z, Li J H, Kim S O.Apoptosis of lung carcinoma cells induced by a flexible optical fiber-based cold microplasma. Biosensors & Bioelectronics, 2011 28: 333-338.
  • Song C H, Attri P, Ku S K, HanI, Bogaerts A, Choi E H. Cocktail of reactive species generated by cold atmospheric plasma: oral administration induces non-small cell lung cancer cell death. Journal of Physics D-Applied Physics, 2021 54: 185202.

Round 2

Reviewer 1 Report

The manuscript is revised, and the quality of the manuscript is increased significantly. 

I noticed only three minor errors. 

Line #96: The resolution of CPI formula is compromised. 

Line #150: CCALU-1 should be CALU-1.

Adding a scale bar to the fluorescence microscope image will be helpful.